METHODS

# IMI-driver: Integrating multi-level gene networks and multi-omics for cancer driver gene identification

**Peiting Shi**[1☯], **Junmin Han**[1☯], **Yinghao Zhang**[1], **Guanpu Li**[1], **Xionghui Zhou**[1,2]*

**1** Hubei Key Laboratory of Agricultural Bioinformatics, College of Informatics, Huazhong Agricultural University, Wuhan, People's Republic of China, **2** Key Laboratory of Smart Farming for Agricultural Animals, Ministry of Agriculture and Rural Affairs, People's Republic of China

☯ These authors contributed equally to this work.
* zhouxionghui@mail.hzau.edu.cn, zhouxionghui6@gmail.com

**Data Availability Statement:** The data and scripts can be downloaded from https://github.com/coding0lion/IMI-driver.git.

**Funding:** This work was supported by Biological Breeding-Major Projects (2023ZD04061), the

## Abstract

The identification of cancer driver genes is crucial for early detection, effective therapy, and precision medicine of cancer. Cancer is caused by the dysregulation of several genes at various levels of regulation. However, current techniques only capture a limited amount of regulatory information, which may hinder their efficacy. In this study, we present IMI-driver, a model that integrates multi-omics data into eight biological networks and applies Multi-view Collaborative Network Embedding to embed the gene regulation information from the biological networks into a low-dimensional vector space to identify cancer drivers. We apply IMI-driver to 29 cancer types from The Cancer Genome Atlas (TCGA) and compare its performance with nine other methods on nine benchmark datasets. IMI-driver outperforms the other methods, demonstrating that multi-level network integration enhances prediction accuracy. We also perform a pan-cancer analysis using the genes identified by IMI-driver, which confirms almost all our selected candidate genes as known or potential drivers. Case studies of the new positive genes suggest their roles in cancer development and progression.

## Author summary

Cancer driver identification is of great value in cancer research. Recent advancements have demonstrated that incorporating the intricate regulatory relationships among genes can facilitate driver genes identification. However, current methods do not fully capture the regulatory relationships at different layers. In this study, we propose a new method for predicting cancer driver genes (IMI-driver), which integrates multi-omics data into eight biological networks (including common networks such as protein-protein interaction networks and various cancer-specific networks) and embeds multi-level network information as features to predict driver gene sets. Experiment demonstrates that our method outperforms mainstream tools, and can discover new drivers.

Fundamental Research Funds for the Central Universities to X. Z. (2662023XXPY003) and the National Training Program of Innovation and Entrepreneurship for Undergraduates of Huazhong Agricultural University (202210504100) to P.S. The funders had no role in study design, data collection and analysis, decision to publish, or preparation of the manuscript. No authors have received a salary from the funders.

**Competing interests:** The authors have declared that no competing interests exist.

## 1 Introduction

Cancer results from critical genetic changes that disrupt the balance between cell proliferation and death [1]. These changes affect cancer driver genes that regulate cell growth, cell cycle, and DNA replication. Finding these genes is essential for early detection, effective therapy, and precision medicine of cancer. However, this is a challenging task due to cancer's complexity and computational and experimental limitations. Large-scale cancer genomics projects, such as the Cancer Genome Atlas (TCGA) [2] and the International Cancer Genome Consortium (ICGC) [3], have produced massive genomic data across various cancer types [4], revealing the causes of cancer. Identifying driver genes that contribute to cancer development is a key challenge in cancer genomics. Different principles have been proposed to distinguish driver genes from passenger genes that accumulate mutations by chance.

One principle involves mutation frequency-based methods, such as DriverML [5] and WITER [6], which assume that high-frequency mutations are indicative of driver genes. However, this principle faces challenges in accurately estimating the background mutation rate [7] and in identifying driver genes with low-frequency or non-coding mutations. In fact, driver genes are not only those that are frequently mutated but also those that play critical roles in cancer development and progression [8], even if their mutation frequencies are relatively low. Another principle focuses on functional impact, as exemplified by methods like MutPanning [9] and OncodriveFML [10]. Machine learning models have been successfully applied to predict key regulatory elements in the genome [11–14]. In this context, such methods utilize machine learning models to predict driver genes based on the functional impact of mutations. While this approach can detect rare mutant genes, it relies on high-quality datasets of both functional and nonfunctional mutations, which are often challenging to obtain [15]. Cancer arises from the dysregulation of multiple cellular mechanisms, including gene expression, protein interactions, and epigenetic modifications, which collectively influence cancer development [16]. Therefore, integrating multi-omics data is crucial for comprehensively identifying potential driver genes, distinguishing them from passenger genes, and providing precise targets for cancer research [17]. A third principle is network analysis, which considers gene interactions and addresses some of the limitations of the previous two principles. Network analysis can identify low-frequency driver genes that mutation frequency-based methods might overlook and can integrate multi-omics data to elucidate the complex mechanisms underlying cancer development.

Cancer progression may result from mutations mediated via protein interactions, illuminated by the protein-protein interaction (PPI) network [18]. Numerous approaches prioritize cancer genes based on PPI networks; some integrate somatic mutation data (e.g., MaxMIF [19], MUFFINN [20], DriverRWH [21]), while others use multi-omics data for network representation learning and model building (e.g., EMOGI [22], MTGCN [23]). Current PPI networks are still incomplete and noisy [24], limiting their ability to capture complex biological relationships and impacting prediction accuracy. Most methods focus on a single PPI network, overlooking interactions across different layers. Constructing a model to integrate multiple networks is crucial for a comprehensive gene interaction landscape. MODIG [25] combines heterogeneous information into multi-dimensional gene networks and integrates multi-omics data. However, it lacks tumor-specific networks, missing tumor-specific regulation information, and doesn't use methylation data, somatic mutations, and other data for constructing comprehensive biological networks representing cancer interactions at various layers.

Here we present IMI-Driver, a novel framework that integrates multi-level network embedding and machine learning to predict cancer driver genes. IMI-Driver constructs various networks based on multi-omics data (gene expression, miRNA expression, DNA methylation,

and somatic mutations), such as gene dependency network (GDN) [26], competing endogenous RNA (ceRNA) network [27], gene co-expression network (GCN) [28], DNA methylation interaction network (DMIN) [29] and co-mutations network (DCMN), to capture the multidimensional interactions of carcinogenesis for each cancer type. It also incorporates other networks that reflect common gene interactions, such as the protein-protein interaction (PPI) network [18], gene pathway similarity network (GPSN) [30], and transcriptional regulatory network (TRN) [31]. Then, it applies Multi-view collAborative Network Embedding (MANE) [32] to extract gene interaction pairs in different networks and embed them into a low-dimensional vector space. In addition, it also integrates biological characteristics extracted from multi-omics data of each gene as features. Finally, it employs XGBoost [33] to build a prediction model based on the embedded vectors. We demonstrate that IMI-Driver outperforms existing methods on several benchmark datasets and reveals some novel driver genes with biological evidence.

## 2 Results

### 2.1 Overall framework

IMI-Driver is a novel method that uses multi-omics data and network features to identify driver genes in various cancers based on the TCGA database. Fig 1 illustrates the IMI-Driver framework. Firstly, we constructed eight biological networks for each cancer by combining gene expression, miRNA expression, DNA methylation, and somatic mutation data with gene regulatory relationships from known databases (Method). These networks include common networks (e.g., PPI, co-pathway network, and TF-target network) and cancer specific networks (e.g., co-expression network, co-methylation network, co-mutation network, ceRNA network, and gene dependency network). Secondly, we extracted three types of relationship pairs: diversity, first-order, and second-order collaboration, from these heterogeneous networks and applied a multiplex heterogeneous network embedding algorithm (MANE) to learn a low-dimensional continuous vector representation for each gene. Thirdly, we enhanced the network features by integrating other omics features such as mutation, genome, epigenetics, and phenotype features (Method) to obtain a comprehensive feature representation for each gene. Finally, an XGBoost model was employed to predict the driver scores for each gene.

### 2.2 Performance assessment of IMI-Driver

We evaluated the performance of IMI-Driver by comparing it with nine representative methods on nine established gold-standard sets of driver genes, comprising five network-based algorithms (DriverRWH [21], MaxMIF [19], DNsum and Dnmax [20], EMOGI [22]), two function-based algorithms (MutPanning [9] and DORGE [34]) and two frequency-based algorithms (WITER [6] and DriverML [5]). We used STRING PPI as the network input for the network-based algorithms [35]. Each method produces a ranked list of individual genes as potential driver genes. Accordingly, we selected the top 100 genes (results for other thresholds are discussed in the "Discussion" section) as potential drivers for each method on nine established gold-standard benchmark datasets (CGC [36], CGCpointMut [35], 20/20Rule [37], HCD [38], oncogene [39], CTAT [40], MouseMut [41], ConsistentSet [35] and IntOGen [42]) (The prediction scores of all genes on nine datasets are available at https://github.com/coding0lion/IMI-Driver.git), and calculated the p-values of the KS-test, MCC, AUROC, F1, and Precision for each method (S1–S4 Figs).

To compare the performance of the different methods, we calculated the performance of each method for each cancer in each benchmark dataset. We will then calculate the average of all benchmark datasets as an overall evaluation of the method. Table 1 shows the ranking of

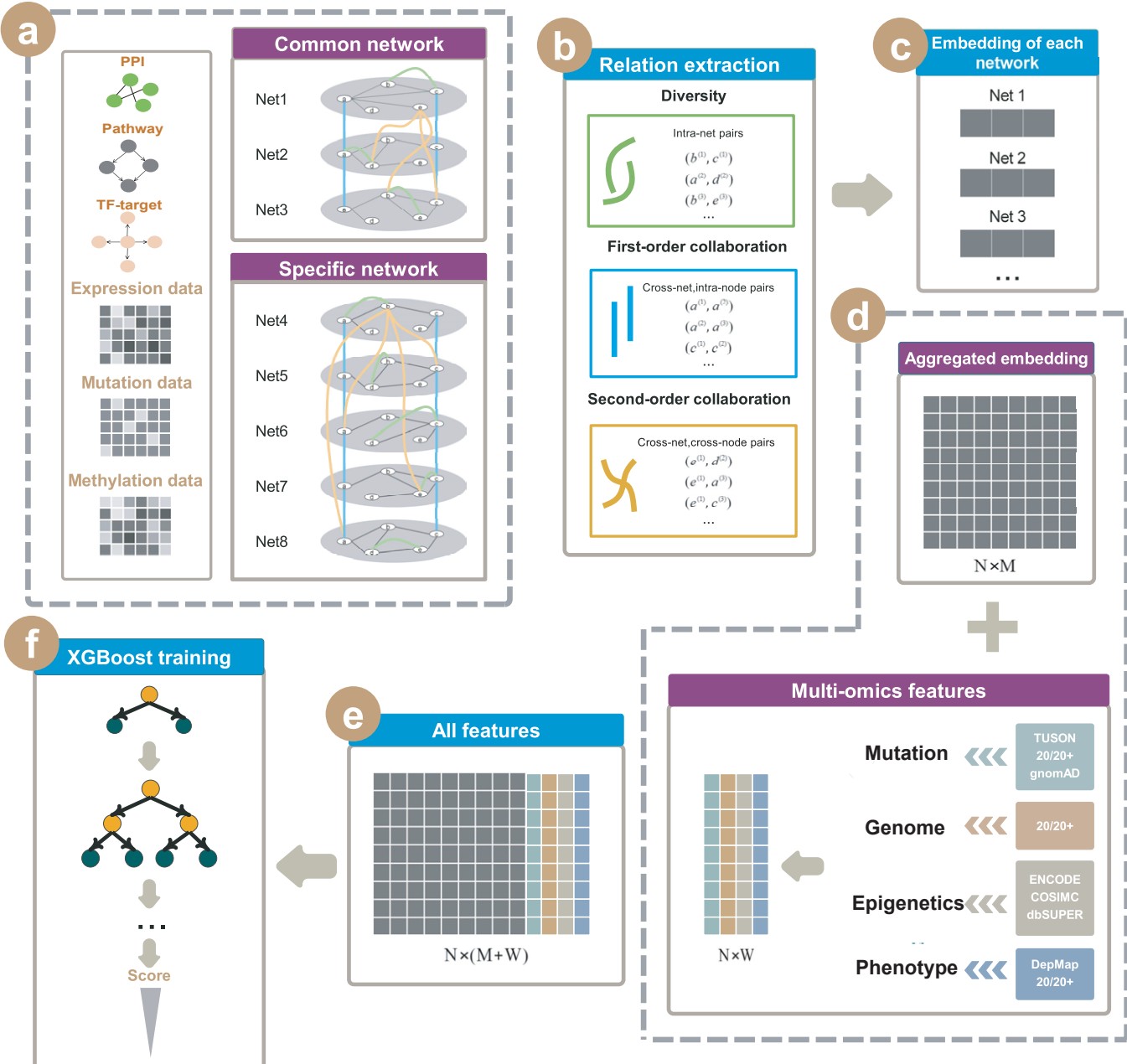

**Fig 1. Overall Framework.** (a) Eight networks were constructed using multi-omics data (five tumor-specific networks) and known regulatory relationships (three common networks). (b) Extracting the eight networks' relation to catch various collaborations between genes. (c) Obtaining the Embedding of each network. (d) Combining the feature in (c) with multi-omics features. (e) All features we generated. (f) Using all features in (e) to train the XGBoost model.

each method's performance among the nine methods. The main evaluation metrics MCC for all methods on the nine benchmark datasets are shown in Fig 2. From both the overall evaluation (S1 Table shows the evaluation scores of each method on different benchmarks) and MCC comparison (Fig 2), our model outperformed other methods on different benchmarks and criteria. Therefore, IMI-driver can identify cancer drivers with high accuracy and robustness.

**Table 1. Performance rankings of all the models on the benchmark datasets (MCC).**

| method | IntOGen | ConsistentSet | CGC | CGCpointMut | CTAT | MouseMut | CD | oncoGene | 20/20Rule | Average Rank |
|---|---|---|---|---|---|---|---|---|---|---|
| DNmax | 9 | 7 | 7 | 7 | 7 | 6 | 7 | 7 | 7 | 7.11 |
| DNsum | 1 | 6 | 6 | 6 | 6 | 5 | 6 | 5 | 6 | 5.22 |
| DORGE | 8 | 2 | 2 | 2 | 2 | 7 | 2 | 4 | 2 | 3.44 |
| DriverML | 7 | 8 | 8 | 8 | 8 | 8 | 8 | 8 | 8 | 7.89 |
| EMOGI | 5 | 3 | 3 | 3 | 3 | 4 | 3 | 6 | 3 | 3.67 |
| driverRWH | 3 | 4 | 4 | 4 | 4 | 2 | 4 | 2 | 4 | 3.44 |
| MaxMIF | 4 | 5 | 5 | 5 | 5 | 3 | 5 | 3 | 5 | 4.44 |
| MutPanning | 6 | 9 | 9 | 9 | 9 | 9 | 9 | 9 | 9 | 8.67 |
| WITER | 10 | 10 | 10 | 10 | 10 | 10 | 10 | 10 | 10 | 10.00 |
| IMI-Driver | 2 | 1 | 1 | 1 | 1 | 1 | 1 | 1 | 1 | 1.11 |

## 2.3 Contribution of each component

Our manuscript leverages multi-omics data, employing eight networks (three common and five tumor-specific, with most specific networks generated using in-house algorithms) to depict cancer regulatory mechanisms. These networks are integrated and embedded to represent genes, contributing to a stable tumor driver gene model. We analyze the substantial contributions of biological networks and assess each feature category's impact on model performance. Validation was performed on 9 benchmark datasets, including IntOGen [42] with tumor-specific driver genes. Results present mean performance across all datasets and

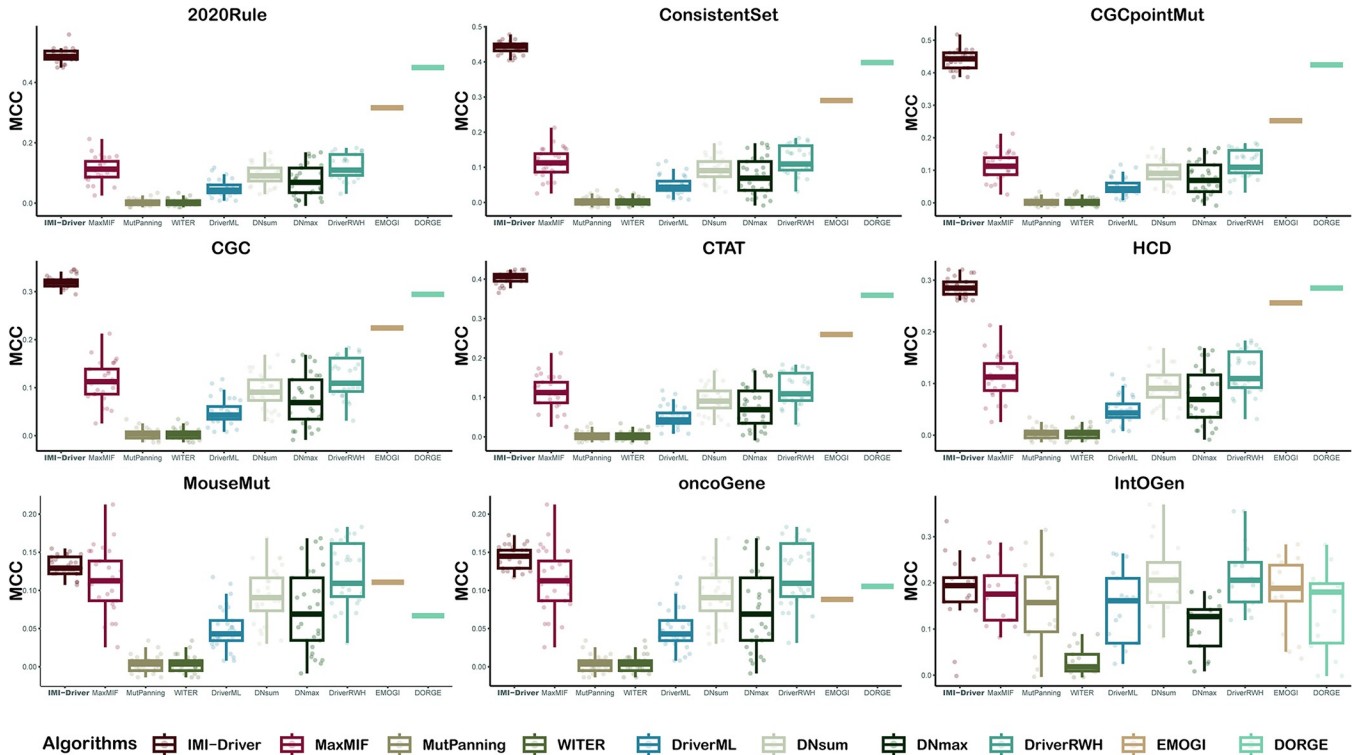

**Fig 2. Performance comparison of IMI-driver with other methods (MCC).** The classification performance of IMI-driver was compared with nine other methods across nine gold-standard datasets. Each subplot shows the classification performance comparison (MCC) of the 10 methods on one benchmark dataset.

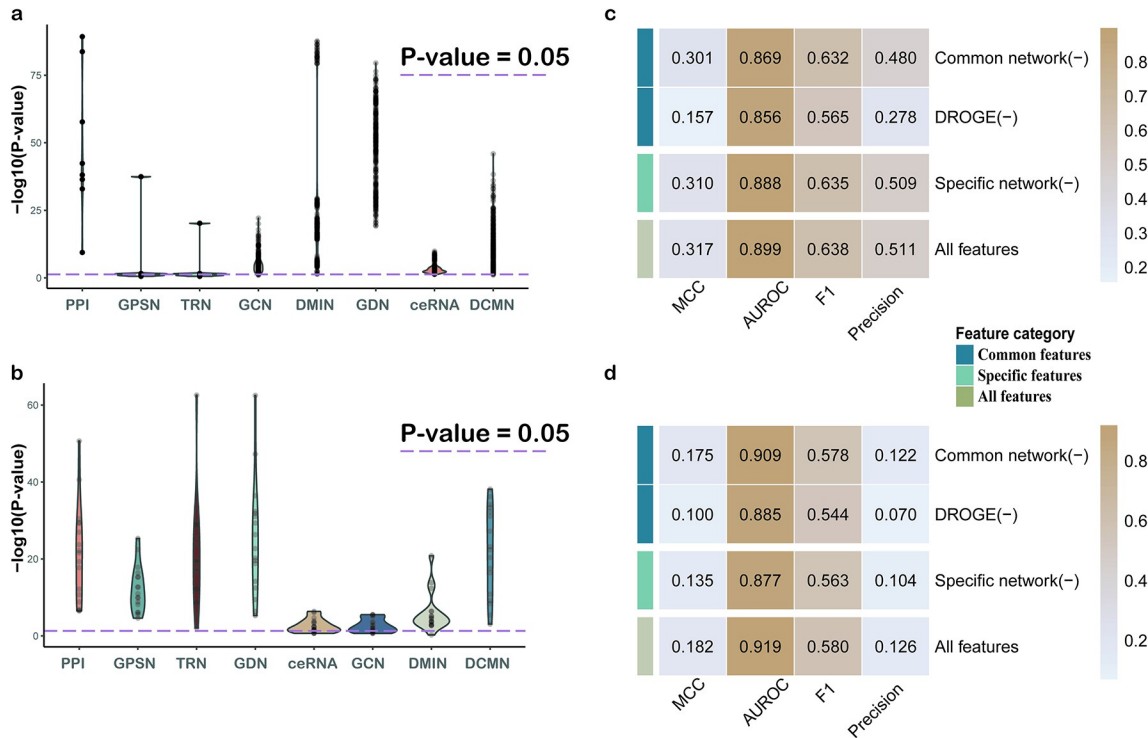

**Fig 3. Contribution of each component in the IMI-driver.** (a) Performance of each network on all benchmarks (each metric is the average across nine benchmark datasets); (b) Performance of each network on IntOGen; (c) Contribution of each component on all benchmarks (each metric is the average across nine benchmark datasets); (d) Contribution of each component on IntOGen.

specifically focus on IntOGen. If a biological network can be utilized for the exploration of cancer driver genes, then these genes should occupy important topological positions within the network. Consequently, we utilized PageRank [43] to rank all genes within each network, followed by conducting enrichment analysis (KS-test) between known driver genes and the ranked gene list to examine whether driver genes are essential nodes within the biological network (i.e., whether they are ranked on the top of the network's gene list ranked by PageRank). The results demonstrate that all networks could facilitate the identification of driver genes (Fig 3A and 3B).

In addition to the eight networks, our method incorporated enhanced features from DORGE [34]. We categorized features into three classes (common network, specific network, DORGE and validated their contributions through ablation experiments. Fig 3 illustrates that removing any feature type decreases predictive performance, confirming their collective contribution. Interestingly, in all benchmark datasets, the specificity network contributes less compared to DORGE (Fig 3C). Yet, when using the cancer-specific driver genes as the benchmark, the cancer-specific network's contribution significantly increases (Fig 3D). Particularly in terms of AUROC, the tumor-specific network is the predominant contributor to global prediction accuracy, affirming its role in revealing regulatory mechanisms in specific cancers and facilitating the exploration of cancer driver genes.

## 2.4 Prediction of novel cancer driver genes

Cancer is a highly heterogeneous disease and a list of cancer-specific driver genes is useful for finding each cancer-specific driver gene, we used IntOGen [42], a list of tumor-specific driver genes with abundant cancer-specific information, to find out new driver genes.

We screened genes that appeared in more than five cancer types (See S1 Text for details of the process) and a total of 93 genes were obtained (Fig 4A and S2 Table). Among these genes, 76 were included in the known driver datasets. Among these 76 genes, there are many well-known cancer driver genes, such as BRCA1, BRCA2, KRAS, and TP53BP1 (Tumor Protein P53 Binding Protein 1). This indicates that our method can identify well-known cancer driver genes. For the remaining 17 genes, DNMT3B, RARB, SMAD5, SMARCA5, SMARCC1 and TGFBRAP1 were predicted only by our method (S2 Table), but various evidence suggests that they have a driving role in cancer development.

Finally, we performed a differential expression analysis of the six genes. For this analysis, only 14 cancer types were used because some cancer data lacked the gene expression data of control samples. The results showed significant expression differences for the six potential drivers (Fig 4B–4G). Notably, DNMT3B was differentially expressed in all cancers.

These results suggest that DNMT3B is the most closely associated novel driver gene with cancer among the six genes. DNMT3B is a DNA methyltransferase that catalyzes de novo methylation of CpG islands in the genome, an epigenetic modification that controls gene expression and genome stability [44]. By altering the methylation status of genes involved in various cellular processes, such as DNA damage response, cell cycle regulation, mismatch repair, cell proliferation, differentiation, and survival, DNMT3B can affect tumorigenesis [44]. For instance, DNMT3B overexpression can result in hypermethylation and silencing of BRCA1, CDKN2A, and MLH1, while DNMT3B deficiency can cause hypomethylation and activation of MYC, CCND1, and EGFR [45]. Therefore, DNMT3B may be a novel driver gene for cancer.

The other five genes have also been reported in some literature to be closely associated with cancer. RARB, functioning as a receptor for retinoic acid, primarily orchestrates gene expression activation [46]. SMAD5, a transcription factor integral to the bone morphogenetic protein (BMP) signaling pathway, governs an array of cellular processes, including differentiation, migration, and apoptosis [47]. SMARCA5, the product of a gene encoding a SWI/SNF-associated chromatin remodeling factor, intricately regulates cell cycle progression, DNA damage repair, and gene expression modulation [48]. Another key contributor, SMARCC1, encoded by a gene pivotal to the SWI/SNF complex, participates in chromatin remodeling, gene transcription, and cellular differentiation [49]. Additionally, TGFBRAP1 plays a pivotal role in modulating the TGF-beta/activin signaling pathway, a pathway intricately linked to tumor development and immune system diseases [50]. This multifaceted portfolio of identified driver genes underscores their significance in diverse cellular pathways and affirms their potential as key players in the intricate landscape of cancer biology.

For each gene, we performed survival analysis across TCGA datasets to assess its prognostic potential. For each cancer dataset, we divided patients into two groups with the same number of samples based on each gene's expression levels (high-expression samples vs. Low-expression samples) and used the log-rank test to evaluate the prognostic risk between the two groups. The results show that, except for SMAD5 and SMARCA5, the remaining four genes have significant prognostic ability in multiple cancers, while SMAD5 and SMARCA5 exhibit marginal statistical significance in survival analysis for some cancer datasets. S3 Table presents the results for all cancer datasets, while Fig 5 shows the results for each gene in a case dataset.

In conclusion, we identified six novel cancer genes that previous computational methods missed, and case studies showed they had roles in cancer development and progression. These novel candidates warrant further functional and cancer genomics studies to confirm their role as cancer drivers.

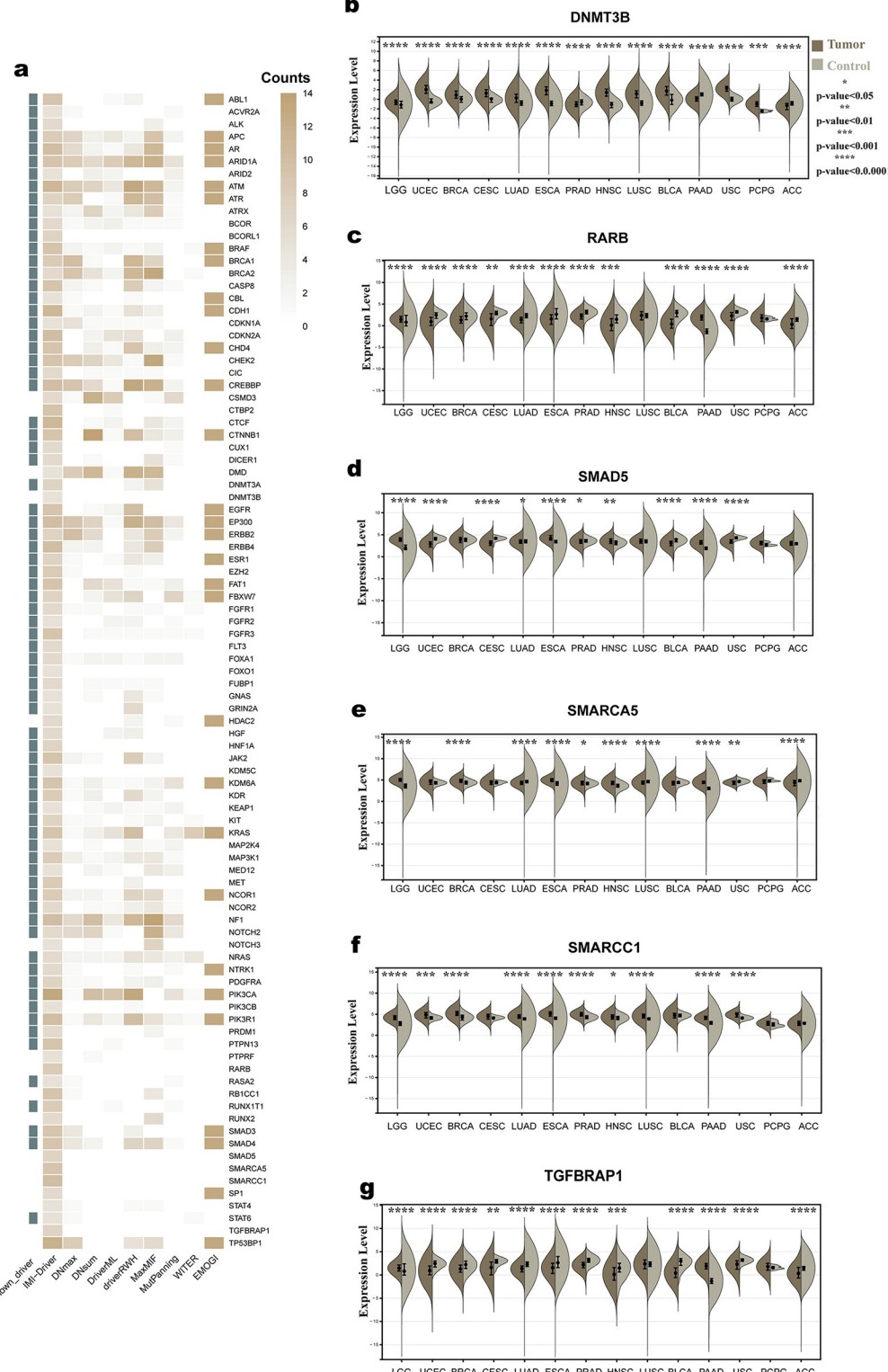

**Fig 4. The top genes identified by IMI-driver and Pan-caner analysis of the novel drivers.** (a) Top genes predicted by IMI-driver; (b-g) Differential expression analysis of the six genes.

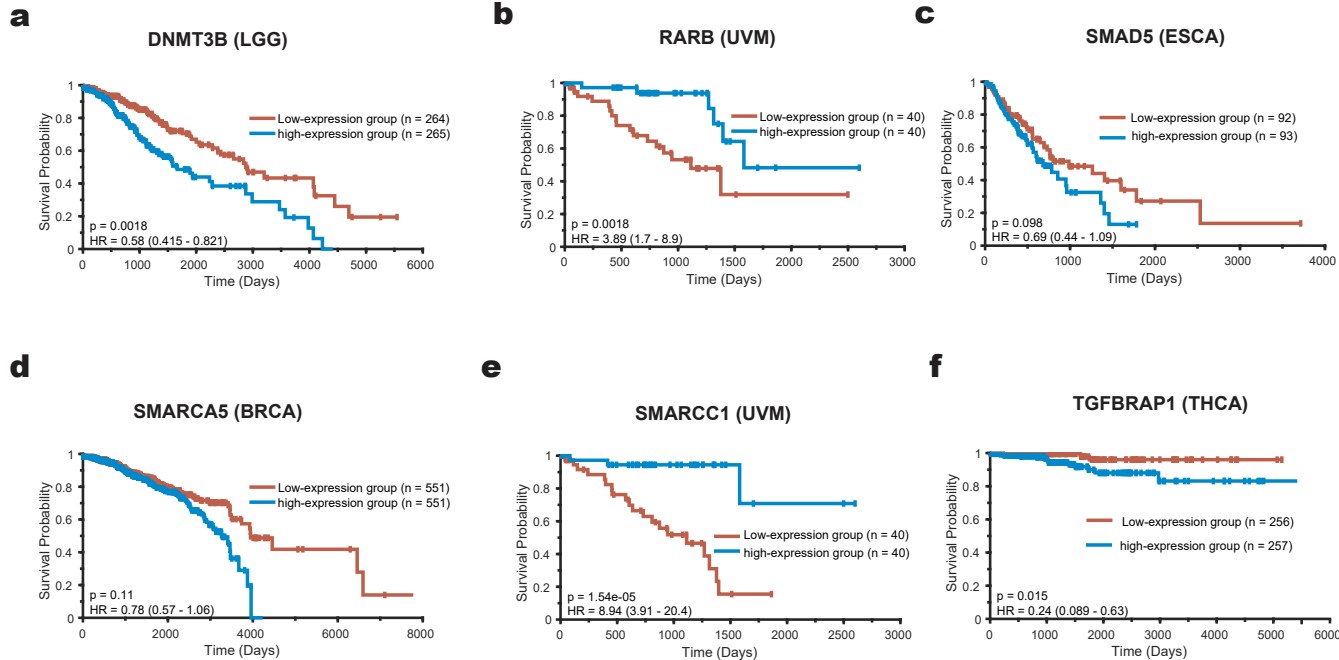

**Fig 5. Survival analysis of the new driver candidates.** In each dataset, patients are divided into two groups based on the expression level of the specific gene (using the median expression value of this gene as the threshold). The log-rank test is then used to assess whether there is a significant difference in prognostic risk between the two groups.

## 3 Discussion and conclusion

Identifying cancer-associated genes that drive the cancer phenotype is a major challenge in cancer genome sequencing. To construct a complete list of cancer driver genes, it is important to integrate high-throughput functional genomics data from diverse sources. Network-based methods can facilitate this integration, but existing methods generally suffer from three problems: (1) Most methods do not make full use of multi-omics data in cancer; (2) most of the methods only used one level of the biological network; (3) The biological networks do not contain cancer-specific regulatory information. We proposed IMI-Driver, a model that integrates cancer-specific and common networks and multi-omics data to predict potential drivers.

IMI-Driver outperformed nine existing tools on nine gold standard gene sets for 29 cancer types from the TCGA, using metrics such as MCC, AUROC, accuracy and KS test. Notably, IMI-Driver also predicted driver genes consistently across different cancer types. Pan-cancer analysis of high-frequency genes predicted as drivers in more than five cancers confirmed most of them as known or potential drivers. Among them, IMI-Driver identified six novel candidates. Differential expression analysis on various cancer datasets and literature review suggests its potential role in the occurrence and progression of cancer.

We assessed the robustness of our method by randomly shuffling gene class labels and rebuilding the prediction model. The results (S5 Fig) indicate that our outstanding performance is not a result of overfitting to noise signals but reflects the identification of crucial drivers in cancer initiation. In the results section, we considered the top 100 genes from each method as driver genes. To evaluate model robustness with changing thresholds (100, 200, 300, 400, and 500), we found that our method consistently outperformed others across nearly all benchmark datasets, demonstrating its robustness (See S6 Fig and S2 Text for details). To evaluate whether the good performance of our method is solely due to the use of the XGBoost

classifier, rather than the effective utilization of multi-omics data and the effective representation of multi-level network information by our model framework, we used several other major classifiers (Support Vector Machine (SVM), Random Forest, Logistic Regression, and Naive Bayes) to construct driver gene prediction models based on the gene representation matrix. The results (S4 Table) show that all these models have relatively good classification performance, especially Logistic Regression, which, as a very simple classifier model, only performs slightly worse than XGBoost. This indicates the effectiveness of our model framework.

In summary, IMI-Driver emerges as a robust and effective model for identifying cancer driver genes, offering scalability to integrate diverse biological networks and datasets. Additionally, we provide a scalable research framework that constructs multi-level networks based on multi-omics data and represents the network nodes for classification. The technical details within the framework can be adjusted, and it can be applied to other biomedical fields beyond driver gene prediction. In this study, we used gene expression data, miRNA expression data, somatic mutation data, and methylation data. Other high-throughput sequencing data, such as proteomics data, can also be incorporated into our framework. At the network level, we integrated eight biological networks, including five cancer-specific networks and three common networks. Of course, other networks, such as metabolic networks, can also be included in the analysis in practical applications. In terms of multi-network embedding, besides the methods we used, other approaches suitable for multi-layer graph embedding can also be applied. For the final model prediction, in addition to driver gene prediction, theoretically, any other type of node class prediction, such as drug target prediction, can also use our framework. In summary, we provide a scalable framework that can be used for various biomedical problems. However, our method has some limitations. For instance, due to the network reconstruction and multi-layer graph embedding calculations, it requires substantial computational resources. For example, using 5 threads (2.90GHz per thread) and 25 GB of memory, our model takes approximately 29 hours. If further reduction in runtime is needed, we recommend using 10 threads and 50GB memory, which can reduce the required time to 22 hours. In future work, we will explore how to optimize our algorithm framework to consume fewer computational resources. Additionally, although our framework can theoretically be applied to other fields, its performance in these areas requires further investigation. This will be a focus of our future work.

## 4 Materials and methods

### 4.1 Data collection

All the data sets we used, including the multi-omics data of 29 cancer types from TCGA, and the known biological network data, were described in S3 Text and S5 Table.

To evaluate the IMI-Driver, an unbiased and comprehensive set of known cancer genes is needed. Unfortunately, there is no such gold standard set of cancer genes, and each cancer gene set is biased toward particular features or study methods. In this work, we used nine cancer driver datasets (see S6 Table for more details) to validate our approach.

1. Cancer Gene Census (CGC; Tier 1; January 2019), a widely used gold-standard set of genes with causal mutations in cancer [36];

2. CGCpointMut, a subset of CGC that contains genes with point mutations in cancer [35];

3. 20/20 Rule, a set of genes with characteristic mutation patterns of oncogenes and tumour suppressors [37];

4. HCD, a consensus set of genes from multiple frequency-based methods [38];

5. oncoGene, a manually curated set of oncogenes from literature and public databases [39];

6. CTAT, an aggregated set of genes from previously reported methods using a combination of tools [40];

7. MouseMut, a set of human homologs of mouse cancer genes identified by insertional mutagenesis [41];

8. ConsistentSet, a non-redundant set of common driver genes from at least three of the above data sets [35];

9. IntOGen [42], a tumour-specific set of driver genes that integrates different prediction methods and provides the best trade-off between sensitivity and specificity. We selected 17 cancers that were shared between IntOGen and TCGA data when this data set was used to validate IMI-driver.

## 4.2 Construction of gene networks

We built eight biological networks from existing studies and classified them into two categories: common networks, using the same data for all cancers; and cancer-specific networks, using the multi-omics data of each cancer type. S4 Text described the three common networks: PPI, gene pathway similarity network, and transcriptional regulatory network. We showed the pipelines of the cancer-specific networks below.

**4.2.1 Gene dependency network.** We used our previous method [26], which calculates the conditional mutual information (CMI) of each gene pair (i.e. gene A and gene B) using their expression data and matched clinical data in cancer patients, to assess gene dependency relationships in each cancer type. The pipeline to construct the gene dependency is shown as follow:

(1) Discretizing the clinical information and gene expression levels. We set the phenotypic status to 1 if distant metastasis occurred within 5 years; if distant metastasis occurred after 5 years, we set the phenotypic status to 0; other data were discarded. The gene expression value was set to 0 if it was below the median gene expression level of all samples; otherwise, it was set to 1.

(2) Inferring the dependency relationship between gene A and gene B by calculating the CMI of each gene pairs.

$$CMI(GeneA; outcome|GeneB) = I_{\text{high}}(GeneA, outcome) - I_{\text{low}}(GeneA, outcome) \qquad (1)$$

Where $I_{\text{high}}$ is the top 35% triplets (i.e. Gene A's value, clinic information, Gene B's value), $I_{\text{low}}$ is the bottom 35% of triplets.

(3) Evaluating the significance of every gene dependency relation using the permutation method.

(4) Constructing a gene dependency network using the significant gene dependency pairs, where the threshold of p-value is 0.05 in this work.

**4.2.2 Competing endogenous RNA network.** Based on the ceRNA hypothesis [51], we screened significant gene pairs sharing the miRNA sponges by using matched gene and miRNA expression data from cancer patients based on our previous method [27]. This method can facilitate biomarker discovery by revealing key ceRNAs that regulate each other through shared miRNAs.

(1) Collecting the miRNA-gene pairs from database targetScan (https://www.targetscan.org/).

(2) Calculating the Pearson correlation coefficient for each miRNA-gene pair using the miRNA and gene expression data, and we will select the negatively correlated pairs with p-values less than 0.05.

(3) Extracting all miRNA from the filtered miRNA-gene pairs.

(4) Testing whether the common miRNA sponges between the two genes were significant using a hypergeometric cumulative distribution function test.

$$p - \text{value} = 1 - F\left(\frac{x}{U}, M, N\right)$$

$$= 1 - \sum_{i=0}^{x-1} \frac{\binom{M}{i}\binom{U-M}{N-i}}{\binom{U}{N}} \tag{2}$$

where $x$ stands for the number of common miRNAs that regulate both of the two genes, $U$ is the number of all the miRNAs in this work, $M$ is the size of the miRNA set and $N$ is the size of the miRNA set.

(5) The patient's gene expression data were used to further confirm the ceRNA relationship, and the positively correlated pairs with p-value<0.05 were retained.

(6) Combining all the ceRNA pairs to construct the ceRNA network.

**4.2.3 Co-expression network and co-methylation network.** Rank-based co-expression network [28] was constructed for each cancer type. For each gene, the top k (k = 4 in this work) genes that were the most correlated ones (Pearson correlation coefficient) for each gene were selected as neighbors in the network. Regarding the co-methylation network, we utilized DNA methylation data to compute the correlation between each pair of genes, employing the same approach as in our previous study [29].

$$r_{ij}^{(1)} = \left|\frac{cov(e_i, e_j)}{\sigma(e_i)\sigma(e_j)}\right| \tag{3}$$

As to the co-methylation network, DNA methylation data was used to calculate the correlation of each gene pair.

**4.2.4 Co-mutations network.** Based on TCGA somatic mutation data, for each cancer type, we calculated the cosine similarity between every gene pair. Then we construct the Somatic mutation network using the rank-based method [28].

$$r_{ij}^{(2)} = \frac{m_i \cdot m_j}{|m_i||m_j|} \tag{4}$$

where $r_{ij}^{(2)}$ is the cosine similarity of the mutation vectors of the $i$-th and $j$-th genes, $m_i$ and $m_j$ are the mutation vectors of the $i$-th and $j$-th genes, respectively.

### 4.3 Integrating multi-level gene network

Different biological networks can capture different aspects of the cancer mechanism. Network representation learning (NRL) can integrate these networks into low-dimensional feature vectors for each node [52], enabling cancer driver gene identification by graph convolutional neural networks (GCNs). However, many cancers have small sample sizes and build networks that are too sparse, which poses a challenge to the training and performance of deep learning models. To overcome this limitation, we used advanced multi-view collaborative network embedding (MANE) [32] to embed multiple heterogeneous networks, which has demonstrated excellent performance on biomedical data. The method captures these three categories of relationships to integrate different biological network information and obtain a low-dimensional and continuous vector representation of each gene (node). In this work, we apply MANE to our study as follows.

Diversity: intra-network pairs. To retain the diversity of each network, each node has a network-specific representation such that two nodes associated in the same network should have similar representations for that network. For each network, we generate topologically associated gene pairs from random walks, by the skipgram model [53] to construct the loss function $L_{\text{Div}}$.

First-order collaboration: cross-network, intra-gene pairs. While different network exhibit diversity, the two network-specific embeddings of the same gene should become similar, by optimizing the loss $L_{C1}$.

Second-order collaboration: cross-network, cross-gene pairs. Gene pairs that interact in one network may also interact in other networks. Thus, we optimize the loss $L_{C2}$.

Based on the losses of the three types of gene pairs, the final loss is obtained.

$$L = L_{\text{Div}} + \alpha \cdot L_{C1} + \beta \cdot L_{C2} \tag{5}$$

where $L_{\text{Div}}$, $L_{C2}$ and $L_{C3}$ are the losses of three types of relationships respectively and $\alpha, \beta \geq 0$ are hyperparameters to control the relative importance among the three components.

Network attention. However, in real-world scenarios, the importance or relevance of each network is often non-uniform and varies from gene to gene. Thus, MANE+ is the algorithm that introduces the neural attention mechanism. We first define a score function s: $U \times V \rightarrow \mathbb{R}$ to compute the similarity between any gene $i$ and any network $n$:

$$s(i, n) = \mathbf{z}_2^{(n)} \cdot \tanh(\mathbf{z}_1^{(n)} \circ \mathbf{f}_i + b_1^{(n)} \mathbf{1}) + b_2^{(n)} \tag{6}$$

where $\mathbf{z}_1^{(n)}, \mathbf{z}_2^{(n)} \in \mathbb{R}^D, b_1^{(n)}, b_2^{(n)} \in \mathbb{R}$ are trainable parameters specific to network $n$, and $\circ$ denotes the Hadamard product. The scores are further normalized to obtain the attention value $a_i^{(n)}$, i.e., the weight of network v w.r.t. gene i:

$$a_i^{(n)} = \frac{\exp s(i, n)}{\sum_{n' \in N} \exp s(i, n')} \tag{7}$$

Subsequently, for every gene $i$, we aggregate its network-specific embeddings into a vector $\mathbf{f}_i^a \in \mathbb{R}^D$, weighted by its network attention:

$$\mathbf{f}_i^a = \tanh(\oplus_{n \in N} a_i^{(n)} \mathbf{f}_i^{(n)}) \tag{8}$$

The attention-based embeddings, $\{\mathbf{f}_i^a : i \in U\}$, can be used as features for downstream applications, where some supervision is often available to guide the learning of network-

specific attention parameters. Thus the following losses in gene classification can be minimized:

$$L_{\text{Att}} = \frac{1}{|U_{\text{train}}|} \sum\nolimits_{i \in U_{\text{train}}} \text{CrossEntropy}\big(y_i, h\big(\mathbf{f}_i^a\big)\big) \qquad (9)$$

where $U_{\text{train}}$ is the set of training genes, $y_i$ is the class of gene $i$, and $h(\mathbf{f}_i^a)$ is the classification output for gene $i$ using the attention-based embedding $\mathbf{f}_i^a$. Since $U_{\text{train}} \subset U$, this is a semi-supervised setting.

The final loss function with the attention mechanism is obtained:

$$L^+ = L_{\text{Div}} + \alpha \cdot L_{\text{C1}} + \beta \cdot L_{\text{C2}} + \gamma \cdot L_{\text{Att}} \qquad (10)$$

where $\gamma > 0$ is a hyperparameter to control the contribution of the attention-based loss.

### 4.4 Feature enhancement

To integrate multifaceted data to predict cancer driver genes, we used DORGE [34]. The 75 features mentioned in the algorithm were used as our source of enhanced features. We selected 45 features from this set, based on KL divergence Ranking [54], for our analysis. See S7 Table for details of feature enhancements.

### 4.5 Assessment criteria

Due to the severe imbalance between positive and negative samples in the dataset, we chose the Matthews Correlation Coefficient (MCC) as the primary evaluation metric. In previous studies, the KS test was used to assess whether a gene set is significantly enriched at the top of a ranked gene list [55]. Following this strategy, we used it to evaluate whether the list of driver genes is significantly enriched in our ranked gene list (where higher ranks indicate a higher likelihood of being predicted as driver genes by our model). In this context, a smaller p-value (from the KS test) indicates a higher degree of enrichment of true driver genes within the model-predicted gene list. Therefore, we have included the KS test along with accuracy, the Receiver Operating Characteristic Curve (AUROC), recall, F1-score, and precision as additional evaluation metrics for the model. S5 Text provides a detailed description of these metrics.

### 4.6 Driver prediction model

In this study, we constructed an XGBoost classifier to predict cancer driver genes using a combination of a network embedding matrix and enhanced features as features. To address the challenge of class imbalance, we employed the SMOTE algorithm [56] to balance the samples in the training set. The key steps of our approach, are detailed in S6 Text. K-fold cross-validation (K = 10 in this work) was applied to validate our model (S7 Text and S7 Fig).

## Supporting information

**S1 Text. Prediction of novel cancer driver genes.**
(DOCX)

**S2 Text. Stability evaluation of IMI-driver.**
(DOCX)

**S3 Text. Data collection.**
(DOC)

**S4 Text. The pipeline of constructing common networks.**
(DOCX)

**S5 Text. Description of assessment criteria.**
(DOCX)

**S6 Text. The pipeline of predicting model.**
(DOCX)

**S7 Text. Details of K-fold cross-validation.**
(DOCX)

**S1 Fig. Performance comparison of ten methods on nine benchmarks (AUROC).** We compared the classification performance of IMI-driver with nine other methods across nine gold-standard datasets. Each subplot shows the classification performance comparison (AUROC) of the 10 methods on one benchmark dataset.
(TIF)

**S2 Fig. Performance comparison of ten methods on nine benchmarks (Precision).** We compared the classification performance of IMI-driver with nine other methods across nine gold-standard datasets. Each subplot shows the classification performance comparison (Precision) of the 10 methods on one benchmark dataset.
(TIF)

**S3 Fig. Performance comparison of ten methods on nine benchmarks (F1-score).** We compared the classification performance of IMI-driver with nine other methods across nine gold-standard datasets. Each subplot shows the classification performance comparison (F1-score) of the 10 methods on one benchmark dataset.
(TIF)

**S4 Fig. Performance comparison of ten methods on nine benchmarks (KS test p-value).** We compared the classification performance of IMI-driver with nine other methods across nine gold-standard datasets. Each subplot shows the classification performance comparison (KS test p-value) of the 10 methods on one benchmark dataset.
(TIF)

**S5 Fig. Comparing the performance of IMI-driver and models based on randomly labeled datasets.** The models were compared across 9 benchmark datasets, with the x-axis representing the models at different thresholds (n) and the y-axis depicting the corresponding results (MCC).
(TIF)

**S6 Fig. Comparing the performance of IMI-driver and other models at various thresholds.** The models were compared across 9 benchmark datasets, with the x-axis representing the models at different thresholds (n) and the y-axis showing the MCC of our method minus the MCC of the best-performing model among other models.
(TIF)

**S7 Fig. K-fold cross-validation.** (a) Diagram of data partitioning for K-fold cross-validation. (b) Diagram of model training and validation for each fold.
(TIF)

**S1 Table. Performance of all the models on the benchmarks.**
(XLSX)

**S2 Table. The high-confidence genes predicted by IMI-driver.**
(XLSX)

**S3 Table. Survival analysis of the potential driver genes on TCGA (Log-rank P-value).**
(XLSX)

**S4 Table. Performance comparison of different classifiers.**
(XLSX)

**S5 Table. 29 cancer types used in this work.**
(XLSX)

**S6 Table. Nine benchmark datasets.**
(XLSX)

**S7 Table. 45 biological features.**
(XLSX)

## Author Contributions

**Conceptualization:** Xionghui Zhou.

**Data curation:** Peiting Shi, Junmin Han, Yinghao Zhang, Guanpu Li.

**Formal analysis:** Peiting Shi, Junmin Han, Yinghao Zhang, Guanpu Li.

**Investigation:** Peiting Shi.

**Methodology:** Peiting Shi, Xionghui Zhou.

**Project administration:** Xionghui Zhou.

**Software:** Peiting Shi.

**Supervision:** Xionghui Zhou.

**Visualization:** Peiting Shi, Junmin Han.

**Writing – original draft:** Peiting Shi, Junmin Han, Xionghui Zhou.

**Writing – review & editing:** Peiting Shi, Junmin Han, Yinghao Zhang, Guanpu Li, Xionghui Zhou.

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
