## [Decision Letter · Decision Letter 0]

13 May 2024

Dear Dr., Zhou,

Thank you very much for submitting your manuscript "IMI-driver: integrating multi-level gene networks and multi-omics for cancer driver gene identification" for consideration at PLOS Computational Biology.

As with all papers reviewed by the journal, your manuscript was reviewed by members of the editorial board and by several independent reviewers. In light of the reviews (below this email), we would like to invite the resubmission of a significantly-revised version that takes into account the reviewers' comments.

We cannot make any decision about publication until we have seen the revised manuscript and your response to the reviewers' comments. Your revised manuscript is also likely to be sent to reviewers for further evaluation.

Sincerely,

Rodrigo Mora

Academic Editor

PLOS Computational Biology

Pedro Mendes

Section Editor

PLOS Computational Biology

Reviewer's Responses to Questions

**Comments to the Authors:**

Reviewer #1: The identification of cancer driver genes is crucial for early detection, effective therapy, and precision medicine of cancer. This study introduces IMI-driver, a novel method that integrates multi-level gene networks and multi-omics data for the identification of cancer driver genes. The authors provide insights into the importance of integrating diverse biological networks (including common networks and tumor-specific networks) and multi-omics data for accurate cancer driver gene identification, and demonstrated the superior performance of IMI-driver compared to existing methods in predicting cancer driver genes.

While IMI-driver presents several strengths in integrating multi-level gene networks and multi-omics data for cancer driver gene identification, the manuscript also have some potential weaknesses need to be improved. My additional comments are as follows:

(1)In this paper, the authors used Top100 genes as the screening threshold. I don't think that's a fair way to compare different methods. As many methods do not use rankings as the screening threshold.

(2)As the field of cancer driver gene identification is rapidly evolving, IMI-driver should compare with the latest state-of-the-art network-based methods, such as MODIG, MTGCN, MNGCL, and so on, to ensure its competitiveness and relevance in the current landscape.

(3)The interpretability of the results and the biological relevance of the identified novel driver genes could be further elaborated to provide a clearer understanding of the implications for cancer research and precision medicine.

(4)The figure legends are too simple in such as Fig. 2 and Supplementary Figures, which don't show the message clearly

(5) The sample number of the multi-omics data of 29 cancer types from TCGA should be provided in the materials and methods, as well as the details resource or version of TCGA.

Reviewer #2: Considering that cancer is caused by the dysregulation of several genes at various levels of regulation, the authors present IMI-driver, a model that integrates multi-omics data into eight biological networks to predict cancer drivers. The experimental results confirm the effectiveness of this method. The major highlight of this work is the integration of different levels of biological networks, especially the incorporation of cancer-specific networks into the gene representation process, which has not been addressed in other methods. Therefore, I believe this work is quite innovative. Additionally, the manuscript is well-structured and written, and the results are promising. However, some points need to be taken into consideration.

Major issues:

1.The authors define the top 100 genes as driver genes and define six new cancer driver genes by comparing them with known driver gene sets. Experimental findings suggest that these genes might indeed be potential driver genes. I am curious whether selecting the top 150 or 200 genes for each cancer type as driver genes would still yield potential driver genes. The authors should demonstrate this result to show the robustness of their method.

2.Regarding Figure 2, the evaluation metrics seem limited, and additional evaluation metrics may be needed.

3.Evaluate the computational efficiency of the IMI-driver model, particularly in terms of runtime and memory usage. Discuss strategies employed to optimize performance, such as parallel processing or algorithmic optimizations, and provide recommendations for researchers seeking to apply the model to large-scale datasets.

4.Discuss avenues for refining the IMI-driver model, such as incorporating additional omics data modalities or exploring alternative network integration techniques. This would guide researchers in extending the proposed methodology and addressing remaining challenges in cancer genomics.

Minor issues:

1.Besides some conventional evaluation metrics, the authors also use the Ks-test to assess the performance of each method. However, the authors did not explain why the Ks-test can serve as an effective evaluation metric.

2.The clarity of Figure 2 is insufficient, especially the text inside it is quite blurry, which affects its readability. I suggest the authors remake Figure 2.

3.There are too few figures in the main text, and more details may need to be presented in the main text.

4.Provide more detailed explanations of the IMI-driver model, including the specific algorithms utilized for multi-view collaborative network embedding and the integration of multi-omics data into biological networks.

Reviewer #3: In this study, the authors propose IMI-driver, a model that integrates multi-omics data into eight biological networks and applies Multi-view Collaborative Network Embedding to embed the gene regulation information from the biological networks into a low dimensional vector space to identify cancer drivers. However, the following issues need to be resolved by the authors before consideration for publication.

1. The authors should give a clear definition of what is cancer driver gene. Regarding the concept of a cancer driver gene, if the author's definition is solely based on genomic mutations, it raises the question of why it is necessary to integrate data from multiple omics. If, however, the impact of mutations on the system is considered, please provide specific experiments to demonstrate the extent of influence that driver mutations identified by IMI-driver methods have on cancer-related biological processes. If the scope of cancer driver genes extends beyond genomic mutations, why are comparative methods predominantly derived from driver mutation approaches? In summary, the fundamental issue is that the authors need to clarify their clear definition of driver gene in this paper, otherwise the various downstream evaluations are meaningless.

2 A single gene usually have multiple methylation sites, and similarly, a gene has multiple sites and types of mutations. Please elucidate specifically how the co-methylation network and co-mutation network are constructed.

3. The authors should provide detailed information on the process of obtaining network-specific embeddings for each gene. Is this a supervised or unsupervised learning approach? If it's the former, how is the label for each gene determined?

**Have the authors made all data and (if applicable) computational code underlying the findings in their manuscript fully available?**

Reviewer #1: Yes

Reviewer #2: Yes

Reviewer #3: Yes

PLOS authors have the option to publish the peer review history of their article (what does this mean?). If published, this will include your full peer review and any attached files.

Reviewer #1: No

Reviewer #2: No

Reviewer #3: No
---

## [Decision Letter · Decision Letter 1]

5 Aug 2024

Dear Dr. Zhou,

We are pleased to inform you that your manuscript 'IMI-driver: integrating multi-level gene networks and multi-omics for cancer driver gene identification' has been provisionally accepted for publication in PLOS Computational Biology.

Best regards,

Rodrigo Mora

Academic Editor

PLOS Computational Biology

Pedro Mendes

Section Editor

PLOS Computational Biology

Reviewer's Responses to Questions

**Comments to the Authors:**

Reviewer #1: The authors have addressed all my concerns. I have no more comments.

Reviewer #2: The authors have adequately addressed all of my questions in this revised version. I have no further concerns or new questions. Based on the authors' responses, I recommend that this paper be accepted.

Reviewer #3: The author's responses and revisions have addressed all of my concerns. I think it is now ready for publication.

**Have the authors made all data and (if applicable) computational code underlying the findings in their manuscript fully available?**

Reviewer #1: Yes

Reviewer #2: Yes

Reviewer #3: Yes

PLOS authors have the option to publish the peer review history of their article (what does this mean?). If published, this will include your full peer review and any attached files.

Reviewer #1: No

Reviewer #2: No

Reviewer #3: **Yes: **Huiyan Sun

---

## [Editor Report · Acceptance letter]

17 Aug 2024

PCOMPBIOL-D-24-00654R1 

IMI-driver: integrating multi-level gene networks and multi-omics for cancer driver gene identification

Dear Dr Zhou,

I am pleased to inform you that your manuscript has been formally accepted for publication in PLOS Computational Biology. Your manuscript is now with our production department and you will be notified of the publication date in due course.

With kind regards,

Zsofia Freund
